# HIV Promotes Atherosclerosis via Circulating Extracellular Vesicle MicroRNAs

**DOI:** 10.3390/ijms24087567

**Published:** 2023-04-20

**Authors:** Andrea Da Fonseca Ferreira, Jianqin Wei, Lukun Zhang, Conrad J. Macon, Bernard Degnan, Dushyantha Jayaweera, Joshua M. Hare, Michael A. Kolber, Michael Bellio, Aisha Khan, Yue Pan, Derek M. Dykxhoorn, Liyong Wang, Chunming Dong

**Affiliations:** 1Interdisciplinary Stem Cell Institute, University of Miami Miller School of Medicine, Miami, FL 33136, USA; 2Department of Medicine, University of Miami Miller School of Medicine, Miami, FL 33136, USA; 3Biostatistics Division, Department of Public Health Sciences, University of Miami Miller School of Medicine, Miami, FL 33136, USA; 4John T. Macdonald Foundation Department of Human Genetics, University of Miami Miller School of Medicine, Miami, FL 33136, USA; 5John P. Hussman Institute for Human Genomics, University of Miami Miller School of Medicine, Miami, FL 33136, USA; 6Section of Cardiology, Department of Medicine, Miami VA Health System, University of Miami, Miami, FL 33146, USA

**Keywords:** HIV, extracellular vesicles, ECFCs, atherosclerosis, aging, miRNA

## Abstract

People living with HIV (PLHIV) are at a higher risk of having cerebrocardiovascular diseases (CVD) compared to HIV negative (HIV^neg^) individuals. The mechanisms underlying this elevated risk remains elusive. We hypothesize that HIV infection results in modified microRNA (miR) content in plasma extracellular vesicles (EVs), which modulates the functionality of vascular repairing cells, i.e., endothelial colony-forming cells (ECFCs) in humans or lineage negative bone marrow cells (lin^−^ BMCs) in mice, and vascular wall cells. PLHIV (N = 74) have increased atherosclerosis and fewer ECFCs than HIV^neg^ individuals (N = 23). Plasma from PLHIV was fractionated into EVs (HIV^posEVs^) and plasma depleted of EVs (HIV PL^depEVs^). HIV^posEVs^, but not HIV PL^depEVs^ or HIV^negEVs^ (EVs from HIV^neg^ individuals), increased atherosclerosis in *apoE*^−/−^ mice, which was accompanied by elevated senescence and impaired functionality of arterial cells and lin^−^ BMCs. Small RNA-seq identified EV-miRs overrepresented in HIV^posEVs^, including let-7b-5p. MSC (mesenchymal stromal cell)-derived tailored EVs (TEVs) loaded with the antagomir for let-7b-5p (miRZip-let-7b) counteracted, while TEVs loaded with let-7b-5p recapitulated the effects of HIV^posEVs^ in vivo. Lin^−^ BMCs overexpressing *Hmga2* (a let-7b-5p target gene) lacking the 3′UTR and as such is resistant to miR-mediated regulation showed protection against HIV^posEVs^-induced changes in lin^−^ BMCs in vitro. Our data provide a mechanism to explain, at least in part, the increased CVD risk seen in PLHIV.

## 1. Introduction

The advent of antiretroviral therapy (ART) has changed the clinical course of HIV infection from a fatal disease to a chronic illness by preventing the development of acquired immunodeficiency syndrome (AIDS) through long-lasting viral suppression [1]. However, ART does not fully eradicate the virus, and people living with HIV (PLHIV) develop non-AIDS-related comorbidities such as diabetes mellitus, respiratory diseases, hepatic diseases, and cerebrocardiovascular disease (CVD) [2,3]. The persistent low-level viremia, detectable only by ultrasensitive PCR, coupled with the chronic state of immune activation and inflammation, represent the most significant factor contributing to the elevated risk for the development of CVD [4]. Indeed, epidemiological studies have shown that PLHIV with sustained undetectable plasma viral loads (VL) are 2–3 times more likely to have atherosclerotic cardiovascular disease (CVD) than HIV negative (HIV^−^) subjects with similar CVD risk factors, including smoking, hypertension, hypercholesterolemia, and diabetes melitus [2,5]. However, the molecular mechanisms that link HIV infection, even when well controlled, with non-AIDS related comorbidities, including CVD, remain poorly understood.

Endothelial cells (ECs) possess anti-thrombotic and anti-inflammatory properties and are critical regulators of vascular wall integrity in response to environmental cues [6]. Endothelial colony-forming cells (ECFCs), formerly known as late endothelial progenitor cells (EPCs), are derived from precursor cells that are activated upon vascular injury to promote angiogenesis, EC rejuvenation, and vascular repair [7]. There is evidence suggesting that early stages of atherogenesis are associated with decreased ECFC functionality and reduced ECFC counts [8]. Interestingly, it has been shown that PLHIV have decreased ECFC numbers and impaired ECFC functionality [9]. Viral infection and ART seem to contribute to endothelial dysfunction through multiple potential mechanisms [10]. We and others have used lineage negative bone marrow cells (lin^−^ BMCs)—a cell fraction enriched for endothelial progenitor cells (EPCs)—as a surrogate for EPCs to study the molecular mechanisms that govern EPC functionality and senescence in animal models [11,12,13]. In previous studies, we have reported microRNA (miR)-mRNA pathways that regulate lin^−^ BMC senescence and functionality (angiogenesis and vascular repair) in aging and hypercholesterolemia-induced atherosclerosis [12,13]. As a major class of gene expression regulators, miRs have been implicated in a variety of pathobiological states, including CVD [14], viral infections (e.g., HIV) [15], and drug exposure [16]. Although miRs are generally unstable in the extracellular environment, their integrity is preserved and their functionality is retained when encapsulated and secreted as a component of extracellular vesicles (EVs) [17]. EVs are lipid bilayer-coated nanosized particles secreted by virtually all cell types [18]. These vesicles have a complex cargo whose composition is dependent on the type and the (patho-)physiological state of the cells from which they were produced and includes nucleic acids (e.g., mRNAs and miRs), proteins, lipids, and signal transduction molecules [19]. EVs exert their effect by being absorbed by acceptor cells and releasing their content into the target cell cytoplasm through complex mechanisms (reviewed in [20]). As such, EVs serve as important vehicles for mediating cell-to-cell communication. EVs have been shown to participate in a multitude of biological functions, including tissue regeneration [21]. Studies indicate that some infectious agents, including HIV-1, can have an impact on the composition of the cargo of circulating plasma EVs [22]. This alteration in the composition of EVs may, in turn, affect the types of signals that the EVs transmit and the effects that EVs have on recipient cells. It is known, for example, that the HIV encoded Viral Nef (negative regulatory factor) protein is present in the plasma EVs of some PLHIV, and its presence could contribute to the chronic inflammation state observed in those individuals [23,24,25].

Given the importance of EVs in cell-to-cell communication, we evaluated the impact that EVs isolated from the plasma of PLHIV (HIV^posEVs^) had on the development of atherosclerosis using a well-established atherogenic mouse model (i.e., *apoE*^−/−^ mice fed a high fat/high cholesterol diet). We found that HIV^posEVs^ accelerated atherosclerosis development, promoted cellular senescence and apoptosis within the vascular wall, and impaired the functionality of lin^−^ BMCs in vivo and vascular ECs in vitro. Furthermore, we identified candidate miRs overrepresented in the HIV^posEVs^ as well as a target gene of the candidate EV-miRs that are, at least partially, responsible for mediating the effects of HIV^posEVs^ on lin^−^ BMCs. Collectively, our results indicate that PLHIV (even in those with well controlled viral loads) produces EVs with altered cargo that promote atherogenesis, providing a plausible mechanism for the increased CVD risk seen in these individuals. 

## 2. Results

### 2.1. PLHIV Have Increased Carotid Intima-Media Thickness (cIMT) and Decreased ECFC Numbers

To study the effects of HIV infection on atherogenesis, we recruited 74 PLHIV on ART with well-controlled viral loads (undetectable HIV RNA by RT-PCR) and 23 HIV^neg^ individuals for this study (Appendix A). To minimize the confounding of other risk factors for atherosclerosis and for circulating ECFC levels, we excluded individuals with a history of myocardial infarction, stroke, diabetes, malignancy, pregnancy, smoking, and illicit drug use to allow us to evaluate the degree of atherosclerosis and ECFC reduction due to long-term HIV infection. PLHIV had a higher atherosclerosis burden, as evidenced by increased cIMT compared to HIV^neg^ individuals (Figure 1A). The number of circulating ECFCs (formerly known as late endothelial progenitor cells or EPCs) was significantly decreased in PLHIV, compared to the HIV^neg^ individuals as measured by CFUs (Figure 1B). These results are consistent with previous work showing increased atherosclerosis burden and decreased ECFCs levels in PLHIV, including those with well controlled viral loads and no differences in traditional risk factors relative to the HIV^neg^ individuals [7]. These data support a role for HIV-specific factors in promoting atherogenesis and reducing vascular repair. 

### 2.2. HIV^posEVs^ Promote Atherogenesis in apoE^−/−^ Mice

To investigate the potential mechanisms underlying the increased atherosclerosis burden seen in PLHIV, we focused on circulating EVs from peripheral blood. To test our hypothesis that EVs from PLHIV carry the signals that modulate atherogenesis, a proatherogenic animal model was used in which 3-week-old *apoE*^−/−^ mice fed a high-fat diet were injected weekly for 12 weeks with: (1) circulating EVs from PLHIV (HIV^posEVs^), (2) PLHIV plasma depleted of EVs (HIV PL^depEVs^), (3) circulating EVs from HIV^neg^ subjects (HIV^negEVs^), or (4) PBS. These mice were sacrificed following 12 weeks of treatment for pathobiological analysis. H&E and Oil Red O (ORO) staining were performed to evaluate the atherosclerosis burden. Mice treated with HIV^posEVs^ had a higher atherosclerosis burden, as measured by both H&E and ORO staining, than animals that were treated with PBS (Figure 2). Interestingly, no increase in atherosclerosis was seen in mice injected with either HIV PL^depEVs^ or HIV^negEVs^ compared to the PBS treated mice. Analysis of cholesterol levels showed that there were no significant differences in total cholesterol, triglycerides, HDL-C, or LDL-C across all mouse groups (HIV^posEVs^, HIV PL^depEVs^, HIV^negEVs^, or PBS). However, HIV^posEVs^-treated mice had increased serum levels of C-reactive protein compared to the other treatment groups (Appendix A). These data indicate that EVs, rather than other fractions of plasma, serve as carriers of the pro-atherosclerotic signals in PLHIV. In addition, HIV^posEVs^ seem to work via factors other than circulating lipids to promote accelerated atherosclerosis in *apoE*^−/−^ mice. 

### 2.3. HIV^posEVs^ Cause Vascular Cell Senescence and Apoptosis In Vivo

To further investigate HIV^posEVs^-mediated atherogenesis, we asked if HIV^posEVs^ treatment in *apoE*^−/−^ mice could cause vascular senescence and apoptosis—cellular processes associated with atherogenesis. We found that there were significantly higher numbers of senescent and apoptotic cells in the aortic wall, including cells in the endothelial layer, from HIV^posEVs^-treated *apoE*^−/−^ mice compared with the aortas isolated from mice treated with HIV PL^depEVs^, HIV^negEVs^, or PBS (Figure 3; enlargements of representative images are available in Appendix A). These results indicate that treatment of the *apoE*^−/−^ mice with HIV^posEVs^, but not HIV PL^depEVs^ or HIV^negEVs^, induces cell senescence and apoptosis in ECs and SMCs in the vascular wall—phenotypic changes associated with accelerated cellular aging that promote atherosclerosis. 

### 2.4. HIV^posEVs^ Affect lin^−^ BMC Senescence and Functionality

To elucidate the mechanisms whereby HIV^posEVs^ mediated the effects of HIV infection on atherogenesis, decreased ECFC levels, and vascular senescence, we examined the effects of HIV^posEVs^ treatment on lin^−^ BMC senescence and functionality. The lin^−^ BMC fraction is enriched for EPCs that contribute to ECFCs, and previous work has demonstrated that lin^−^ BMCs promote vascular repair and angiogenesis [12,13,26]. Similar to what was seen with the vascular cells, we observed a relatively high percentage of β-gal positive lin^−^ BMCs in the PBS control group, which is consistent with premature aging seen in *apoE*^−/−^ mice [27]. Treatment with HIV^posEVs^ further accelerated lin^−^ BMC senescence (Figure 4A,B). There was no significant difference in the percentage of β-gal positive cells in the HIV PL^depEVs^ and HIV^negEVs^ treatment groups compared to the PBS control treatment. In addition, lin^−^ BMCs from the HIV^posEVs^-treated animals had decreased migratory capacity, as determined by the wound healing (scratch) assay (Figure 4C,D), as well as diminished proliferation, as measured by the MTT assay (Figure 4E). This decreased migratory capacity and proliferation were not seen in the HIV^negEVs^ or HIV PL^depEVs^ treatment groups. To further assess the impact of HIV^posEVs^ on lin^−^ BMC health and functionality, lin^−^ BMCs isolated from 3-week old *apoE*^−/−^ mice were cultured in vitro in the presence of HIV^posEVs^, HIV PL^depEVs^, HIV^negEVs^, or PBS. The in vitro treatment of lin^−^ BMCs with HIV^posEVs^ for 96 h showed significantly elevated levels of cell senescence compared to lin^−^ BMCs treated with either HIV PL^depEVs^, HIV^negEVs^, or PBS, as measured by β-gal staining (Figure 4F,G). Therefore, both in vivo and in vitro experiments support that exposure to HIV^posEVs^, but not HIV PL^depEVs^ or HIV^negEVs^, increases lin^−^ BMC senescence with decreased cellular functionality. Since at least some of the ECFCs are believed to be derived from the EPCs encompassed in lin^−^ BMCs, it is reasonable to assume that HIV^posEVs^-induced lin^−^ BMC senescence may have contributed to the decreased ECFCs in PLHIV and the vascular aging process in *apoE*^−/−^ mice treated with HIV^posEVs^. 

### 2.5. HIV^posEVs^ Impairs EC Functionality

Next, we determined if mature vascular ECs would also be affected in the same manner as lin^−^ BMCs by HIV^posEVs^. To that end, we cultured ECs isolated from 3-week-old *apoE*^−/−^ mice for 96 h with HIV^posEVs^, HIV PL^depEVs^, HIV^negEVs^, or PBS. Similar to lin^−^ BMCs, ECs treated with HIV^posEVs^, but not HIV PL^depEVs^ or HIV^negEVs^, showed impaired migration (Figure 5A,B) and reduced proliferation (Figure 5C) compared to PBS-treated ECs. These results indicate that HIV^posEVs^ not only affect lin^−^ BMC health and functionality but also directly target ECs. It is plausible that the combined action of the HIV^posEVs^ on lin^−^ BMCs and ECs may have contributed to the accelerated vascular senescence, injury, and atherogenesis. 

### 2.6. HIV^posEVs^ and HIV^negEVs^ Show Similar Physical Characteristics and Surface Markers

Our data support the role of circulating EVs in mediating, at least in part, the effects of HIV infection on atherosclerosis. To investigate the effect of HIV infection on circulating EVs, we first compared the physical properties of EVs isolated from PLHIV and HIV^neg^ individuals using the Nanoparticle Tracking Analyses (NTA). EVs from both groups showed similar biophysical characteristics, including size, protein concentration, as well as particle/protein, particle/RNA, and particle/lipid ratios (Appendix A). The absence of multiple peaks in the size distribution histograms suggests that both HIV^posEVs^ and HIV^negEVs^ are composed of a uniform distribution of vesicle sizes (Appendix A). Staining of the HIV^posEVs^ and HIV^negEVs^ with antibodies for EV surface markers—CD63, CD81, CD9, and Hsp70—showed that EVs from both groups contained similar levels of these well-established EV surface markers as measured by flow cytometry (Appendix A). To evaluate if the EVs prepared from the plasma of PLHIV were not contaminated by viral particles, HIV^posEVs^ and HIV^negEVs^ samples were analyzed for HIV p24 levels by ELISA. There was minimal p24 expression in both groups, and there was no difference between groups (Appendix A). To evaluate if the EV preparations were contaminated with lipoproteins that could influence the atherosclerotic phenotype, we assessed ApoB 100 and ApoA1 levels by Western blot analysis. This analysis showed that there was no lipoprotein contamination in the EVs (Appendix A). These results indicate that HIV infection does not significantly alter the biophysical characteristics and surface marker expression patterns of plasma EVs.

### 2.7. HIV^posEVs^ Have Differential microRNA Cargo Compared to HIV^negEVs^

Next, we examined whether HIV infection altered the composition of EV cargo, which, in turn, could mediate the effects of HIV infection on atherosclerosis. Since we have previously shown that miRs play important regulatory roles in lin^−^ BMC senescence and CVD [12,13,28], we compared the miR profiles in EVs from PLHIV with the largest plaque areas (N = 5) and HIV^neg^ subjects with no plaques (N = 5), representing the extremes of the disease spectrum. All individuals were male, non-smokers, and had an average age of 47 and 46 years, respectively. Overall, 415 mature miRs were shown to be present in the EVs in both groups. Among these miRs, 126 miRs were upregulated and 49 miRs were downregulated by at least 2-fold in HIV^posEVs^ relative to HIV^negEVs^ (Appendix A). Among the 126 upregulated miRs in HIV^posEVs^, 43 miRs had ≥100 counts per million reads mapped (CPM) in the HIV^posEVs^ samples. Of these 43 miRNAs, let-7b-5p, miR-16-5p, miR-423-5p, miR-103a-3p, and miR-107 were the most abundant miRs in the HIV^posEVs^. The average CPM of these five miRs in each group is shown in Appendix A. Validation of differential expression of these miRs in HIV^posEVs^ versus HIV^negEVs^ was performed using qRT-PCR in an independent sample set (N = 5 samples in each group). Four of these miRNAs: let-7b-5p, miR-16-5p, miR-423-5p, and miR-103a-3p, were validated by qRT-PCR (Appendix A). Putative mRNA targets were identified for each of the 4 validated EV-miRs using Ingenuity Pathway Analysis Software (IPA) (Qiagen) (Appendix A). This analysis showed that the target genes of the four validated EV-miRs were enriched in biological processes involved in cardiac enlargement, cardiac fibrosis, and cardiac cell necrosis/death (Appendix A). A review of literature on these miRs shows that the predicted targets of Let-7b-5p are involved in cardiovascular pathologies such as cardiac fibrosis, dilated cardiomyopathy (DCM), myocardial infarction (MI), arrhythmia, angiogenesis, atherosclerosis, and hypertension, while the targets of miR-16-5p and miR-103a-3p are involved in EPC senescence and impairment of cellular functions, such as proliferation (Appendix A). Collectively, these results indicate that HIV^posEVs^ have higher levels of miRs that could be involved in molecular pathways that contribute to atherosclerosis and lin^−^ BMC dysfunction. These analyses suggest that the candidate EV-miRs had the potential to mediate the effects of HIV^posEVs^ and hence HIV infection, on the cardiovascular system. 

### 2.8. TEVs Overexpressing Antagomirs for Candidate EV-miRs Attenuate the HIV^posEVs^ Effects

To determine if the elevated EV-miRs in PLHIV mediated the effects of HIV^posEVs^ and, therefore, HIV infection on atherogenesis, we examined whether antagonizing these EV-miRs, starting with let-7b-5p and miR-103a-3p, would abrogate the effects of HIV^posEVs^ on lin^−^ BMC senescence. To that end, 3-month-old *apoE^−/−^* mouse MSCs were transduced with miR inhibitors (antagomirs) for miR- let-7b-5p or 103a-3p according to our previously published approach [12]. This approach produced MSC-secreted EVs that were loaded with antagomirs for let-7b-5p (miRZip-Let7b-5p) or miR-103a-3p (miRZip-103a-3p) and were termed tailored EVs (TEVs). Lin^−^ BMCs isolated from 3-month-old *apoE^−/−^* mice were exposed to (1) HIV^posEVs^, (2) PBS, (3) HIV^posEVs^ and miRZip-Let-7b-5p TEVs, (4) HIV^posEVs^ plus miRZip-103a-3p TEVs, or (5) HIV^posEVs^ plus miRZip-Ctr TEVs. As expected, lin^−^ BMCs incubated with HIV^posEVs^ and HIV^posEVs^ plus miRZip-Ctrl TEVs showed significantly increased percentages of β-gal positive cells compared to the PBS-treated samples. In contrast, exposure of lin^−^ BMCs to HIV^posEVs^ in the presence of either the miRZip-Let-7b TEVs or miRZip-103a-3p TEVs abrogated, at least in part, the elevation of β-gal staining seen in the HIV^posEVs^ alone and HIV^posEVs^ + miRZip Ctr treated cells (Figure 6A,B). These results indicate that inhibition of miR-103a-3p or Let-7b-5p could ameliorate the negative effects of HIV^posEVs^/HIV infection on mouse lin^−^ BMC senescence.

### 2.9. Overexpression of Hmga2 Rescues the Effects of HIV^posEVs^ on lin^−^ BMCs

To understand the molecular mechanisms/pathways underlying the effects of EV-miRs on CVD, we focused on let-7b-5p and its downstream target, High Mobility Group AT-Hook 2 (Hmga2) [29]. The let-7b-5p—Hmga2 pathway was chosen because (1) we have previously identified Hmga2 as a key regulator of lin^−^ BMC senescence in the context of aging where it works downstream of miR-21 and miR-10A* [28], (2) the let-7b-5p—Hmga2 axis has been previously shown to regulate neural stem cell senescence, and (3) Hmga2 is the top in silico predicted target for let-7b-5p according to TargetScan with 7 predicted let-7b-5p binding sites in its 3′UTR [29,30]. We used lentiviral constructs to overexpress either wild type (wt) Hmga2 or a miR-resistant form of Hmga2 lacking the 3′UTR (3′UTRdel) in lin^−^ BMCs isolated from 3-month-old *apoE*^−/−^ mice. Stable cell lines were established according to standard protocols. Control transduced (lin^−^ BMCs Ctrl), wt Hmga2, and Hmga2-3′UTRdel-overexpressing lin^−^ BMCs were then incubated with HIV^posEVs^, EVs overexpressing let-7b-5p (let-7b-5p EVs), miRZip-let-7b-5p TEVs, or miRZip-Ctrl EVs. Let-7b-5p EV treatment led to increased senescence in control transduced lin^−^ BMCs compared to control EVs, partially recapitulating the effects of HIV^posEVs^ on lin^−^ BMC senescence (Figure 7). Interestingly, the effects of let-7b-5p EVs on β-gal expression were partially blocked by wt Hmga2 overexpression, whereas Hmga2-3′UTRdel overexpression almost completely blocked the effects of let-7b-5p EVs, similar to that seen with miRZip-let-7b-5p treatment (Figure 7; representative images for the β-gal staining are available in Appendix A). Furthermore, the effects of HIV^posEVs^ were also blunted in Hmga2-3′UTRdel transduced lin^−^ BMCs. These results indicate that the effects of HIV^posEVs^ on lin^−^ BMC senescence is mediated, at least in part, through the engagement of the let-7b-5p—Hmga2 axis. 

## 3. Discussion

It is well documented that PLHIV have an elevated risk of developing a wide range of pathologies, including CVD [31]. Several mechanisms have been implicated in the development of CVD in PLHIV, including oxidative stress, ER stress, NLRP3 inflammasome activation, and autophagy inhibition (reviewed in [32]). Indeed, CVD and non-AIDS-defining cancers are the main causes of morbidity and mortality for PLHIV, particularly in the US, where most PLHIV achieve adequate viral suppression [33,34]. We recruited PLHIV and HIV^neg^ individuals with similar cardiovascular risk factors, and excluded factors that could affect ECFC levels, including malignancy, pregnancy, smoking, and illicit drug use, to determine the atherosclerosis burden and ECFC reduction due to long term HIV infection. We found that cIMT measurement, the surrogate marker for atherosclerosis burden, was significantly increased in PLHIV compared with HIV^neg^ individuals. Atherosclerosis is an inflammatory process that develops in response to endothelial injury [35,36]. Multiple lines of evidence indicate that ECFCs play an important role in repairing injured vessels and in atherogenesis [8]. Conflicting data, however, exist regarding ECFC changes in PLHIV and their role in HIV-associated CVD development [37]. In this study, we report reduced ECFC levels in PLHIV compared to HIV^neg^ individuals, supporting the role of ECFCs in HIV-associated atherosclerosis. 

Several factors, including HIV infection itself, chronic inflammation, and persistent immune activation, are implicated in the increased CVD risk seen in PLHIV [31]. Evidence suggests that sustained monocyte/macrophage activation is a key factor in chronic inflammation that would favor the development of HIV-associated atherogenesis [38]. We hypothesized that CVD risk factors, particularly HIV-specific factors, affect the composition of the cargo of plasma EVs, specifically their miR content. This alteration in EV-miR composition, in turn, promotes the development of CVD. Virtually all cells in the body secrete EVs, and they are present in all biofluids. However, circulating blood cells and endothelial cells lining the inner lumen of the blood vessels are directly affected by the low-level HIV viremia and/or chronic inflammation/immune activation seen in PLHIV, including those with well-controlled viral loads [33,34]. As a result, they are more likely to produce EVs with modified characteristics/cargo reflecting the persistent inflammatory state associated with HIV infection [35]. Thus, analysis of plasma EVs may provide important mechanistic insights about how HIV infection increases the risk for atherogenesis. To study the effects of HIV^posEVs^ in mediating atherosclerosis development, we chose the well-established *apoE*^−/−^ mouse model fed a high fat/high cholesterol diet. Injection of HIV^posEVs^ exacerbated atherosclerosis development in these mice. In contrast, HIV PL^depEVs^ or HIV^negEVs^ had no effect on atherosclerosis burden, showing similar plaque sizes to those seen in the PBS treated mice. These results show, for the first time, that the effects of HIV infection on CVD are mediated, at least in part, by EVs present in the peripheral blood. This model of HIV^posEVs^/*apoE*^−/−^ mice also represents a novel model system to study atherosclerosis associated with HIV infection. The use of HIV^posEVs^ and HIV PL^depEVs^ allowed us to effectively determine the impact of the factors encompassed in the plasma on atherogenesis. The observed differences in atherosclerosis burden between HIV^posEVs^ -and HIV^negEVs^-treated *apoE*^−/−^ mice, together with the lack of differences in atherosclerosis burden between HIV PL^depEVs^- and PBS-treated *apoE*^−/−^ mice, suggest that EVs, rather than other factors in the plasma such as lipid proteins, serve as the primary carriers for the atherogenic signals associated with HIV infection. The similar lipid protein profile between HIV^posEVs^- and HIV^negEVs^-treated *apoE*^−/−^ mice (with different atherosclerosis burdens) also suggests that cholesterol levels are not a determining factor for the increased atherogenesis associated with HIV^posEVs^ injection in *apoE*^−/−^ mice. 

To determine the potential mechanisms underlying the effects of HIV^posEVs^/HIV infection on atherosclerosis, we focused on ECFCs, lin^−^ BMCs, and vascular wall cells. Senescence and dysfunction of ECFCs have been associated with atherosclerosis initiation and progression. Furthermore, injection of BM cells ameliorated atherosclerosis development in *apoE*^−/−^ mice [36]. Several groups, including our laboratory, have shown that lin^−^ BMCs can induce angiogenesis and vascular repair [12,13]. In this study, we have observed that injection of HIV^posEVs^ into *apoE*^−/−^ mice resulted in lin^−^ BMC senescence, as well as impaired proliferation and migration, whereas HIV PL^depEVs^ and HIV^negEVs^ showed no discernible effects on these cells compared to PBS treated mice. β-galactosidase and TUNEL staining revealed increased cellular senescence and apoptosis in the aortas from mice treated with HIV^posEVs^ compared with mice treated with HIV PL^depEVs^, HIV^negEVs^, or PBS. We also studied the effects of HIV^posEVs^ on lin^−^ BMCs and ECs in vitro. Similar to the in vivo observations, exposure of lin^−^ BMCs in vitro to HIV^posEVs^, but not HIV PL^depEVs^ or HIV^negEVs^, resulted in elevated lin^−^ BMC senescence. Furthermore, HIV^posEVs^ caused decreased EC migration and proliferation in vitro. Thus, the increased vascular cell senescence and apoptosis observed in the *apoE*^−/−^ mice treated with HIV^posEVs^ may be the result of both deficient vascular repair and direct insult to vascular cells. The lack of differences in the levels of HDL-C and LDL-C cholesterol levels between *apoE*^−/−^ mice treated with HIV^posEVs^ and controls (Appendix A) provide further support for lin^−^ BMC and vascular cell senescence and apoptosis as possible mechanisms, instead of cholesterol levels, that underlie the effects of HIV infection-specific factors on CVD mediated by HIV^posEVs^. 

To study how HIV infection impacted EVs, we first studied the biophysical properties and EV surface markers and found no significant differences between the two groups [39]. We then focused on the cargo content by specifically analyzing the miR composition of the EVs. Each miR has the potential to regulate multiple target genes and pathways, and in that way exerting greater biological effects than individual proteins and mRNAs. Furthermore, we have previously demonstrated that miR dysregulation plays a key role in lin^−^ BMC senescence, functional impairment, and their angiogenic potential [12,13,28]. Comprehensive analyses and prioritization of the sRNA-seq data obtained in HIV^posEVs^ and HIV^negEVs^ led to the identification and validation of four EV-miRs that are elevated in HIV^posEVs^ relative to HIV^negEVs^ (≥100 copies and ≥2-fold different between groups). Importantly, these EV-miRs and their target genes are implicated in a variety of pathobiological processes consistent with CVD development, including cell senescence, DNA damage, cellular processes, chronic inflammation, hypertension, cardiomyopathy, and atherosclerosis. 

To investigate the role of these candidate EV-miRs in regulating lin^−^ BMC and vascular cell senescence and in affecting the cardiovascular system, we selected two of candidates for further analysis: let-7b-5p and miR-103a-3p. Let-7b-5p has been shown to regulate neural stem cell (NSC) senescence, has been found in myocardial infarction, and is associated with pulmonary arterial hypertension [40,41,42]. MiR-103a-3p has been previously shown to be upregulated in senescent BM stem cells [43]. Furthermore, circulating miR-103a-3p has been shown to contribute to angiotensin II-induced renal inflammation and fibrosis via a SNRK/NF-κB/p65 regulatory axis [44]. Using the approach of generating tailored EVs (TEVs) through genetic modification of MSCs first described by our group [12], we produced TEVs loaded with individual antagomirs for let-7b-5p or miR-103a-3p. Remarkably, these miRZip-let-7b-5p and miRZip-103a-3p TEVs were able, at least partially, to abrogate the effects of HIV^posEVs^ in inducing lin^−^ BMC senescence. 

Hmga2 is a target of let-7b-5p, and the let-7b—Hmga2 axis regulates NSC senescence [45,46,47,48] and has been implicated in cell apoptosis in Parkinson’s disease [29]. Furthermore, we have previously demonstrated that endogenous miR-10A* and miR-21 regulate lin^−^ BMC senescence via targeting Hmga2 [28]. Therefore, we asked if Hmga2 mediated the effects of upregulated let-7b-5p in HIV^posEVs^ on lin^−^ BMCs. We overexpressed wt Hmga2 and Hmga2-3′UTRdel in lin^−^ BMCs. Hmga2-3′UTRdel overexpression partially blocked the effects of HIV^posEVs^ on the senescence of lin^−^ BMCs—an effect that is similar to that seen in lin^−^ BMCs treated with miRZip-let-7b-5p TEVs. Furthermore, Hmga2-3′UTRdel overexpression completely blocked the effects of let-7b-5p overexpress EVs. These results indicate that the overexpressed let-7b-5p in HIV^posEVs^ acts through Hmga2 to mediate, in part, the effects of HIV^posEVs^ in lin^−^ BMCs.

Due to logistical difficulties, we used plasma EVs isolated from previously frozen plasma samples. Freezing and thawing plasma samples could activate platelet and increase the chance of contamination of EVs by EV-like particles produced by platelets activation [49]. However, all samples used in this study from both groups (PLHIV and HIV negative) underwent the same treatment and storage procedures. Therefore, the difference between the effects of HIV^negEVs^ and HIV^posEVs^ could not be accounted for by contamination from EV-like particles produced by platelet activation. To address any potential impact of freezing and thawing on EVs, we performed thorough characterization of the plasma EVs used in this study for their protein membrane profile per the guidelines of the International Society for Extracellular Vesicles (ISEV) [39]. We further analyzed the presence of platelet-derived growth factor (PDGF), which is expressed by platelets, in the plasma EV preparations), and were unable to detect any evidence of levels of PDGF in our EV preparations, arguing against contamination of the EVs by platelet activation. 

Our results are consistent with a previous study showing that plasma EVs from PLHIV with controlled viral loads induce necrosis in HUVECs [50]. In this work, we take an additional step in exploring the possible molecular mechanism underlying the effects of plasma EVs derived from PLHIV on vascular endothelial cells. Our current work focused on plasma EVs-miRNAs and identified several candidate EV-miRNAs that are overexpressed in EVs from PLHIV that account for, partially, the effect of HIV^posEVs^ on atherogenesis. In addition to EV-miRNAs, other EV cargos have been investigated by others. Recently, it has been reported that HIV virally encoded proteins, e.g., nef, can be detected in EVs from bronchoalveolar lavage (BAL) fluid in patients with undetectable viral loads [51]. The Nef carrying EVs induces endothelial cell apoptosis. As such, it is imperative to further investigate different classes of molecules in future studies to fully elucidate the mechanisms at play in the effects of PLHIV EVs on the cardiovascular system.

Although the results presented here show that miRs derived from EVs play a central role in mediating the effect of HIV infection and ART on the development of CVD, we can not rule out that additional molecules delivered by the EVs may also contribute to the development of atherosclerosis in PLHIV. In addition to miRs, EVs contain long non-coding RNAs, mRNAs, proteins, and other classes of molecules that make up EV cargo. In addition, we have shown that PLHIV have decreased numbers of ECFCs compared to HIV negative individuals with similar CVD risk factors. However, we have not shown that this decrease in ECFCs is due directly to exposure to EVs. In future work, we will assess the impact of EVs derived from PLHIV on human ECFCs from uninfected individuals. This will confirm that human ECFCs respond in the same fashion as mouse lin^−^ BMCs to HIV^posEVs^. Furthermore, a comparison between the plasma EVs-miRs found in PLHIV with atherosclerotic plaques and the plasma EVs-miRs from HIV^neg^ individuals with atherosclerotic plaques could refine our understanding of which EV-miRs are specific for CVD development in PLHIV. 

In summary, we have shown that PLHIV have an increased atherosclerosis burden and decreased ECFC levels, relative to HIV^neg^ subjects. We have demonstrated that HIV^posEVs^ recapitulate the effects of HIV infection on atherosclerosis with accompanied changes in lin^−^ BMC senescence and functionality in *apoE*^−/−^ mice. Importantly, we have identified candidate EV-miRs and the involvement of the let-7b-5p—Hmga2 axis in mediating the HIV effects. Experiments are ongoing in our laboratory to delineate the molecular pathways used by other candidate EV-miRs that underlie the HIV effects, with the goal of identifying therapeutic targets that may prevent the exacerbated atherosclerosis observed in PLHIV. 

## 4. Materials and Methods

Laboratory health and safety procedures have been complied with throughout the experimental procedures as described.

### 4.1. Human Subjects

Human subjects who participated in this study were recruited following a protocol approved by the Institutional Review Board (IRB) of the University of Miami (IRB protocol #20140333). All participants gave informed consent prior to their inclusion in this study. PLHIV on ART and HIV^neg^ individuals over the age of 18, from both sexes, were enrolled. Exclusion criteria were a history of CVD events (myocardial infarction and stroke), diabetes, malignancy, pregnancy, smoking (conditions shown to affect ECFC levels and functions), or individuals who lacked the capacity to consent. High-resolution B-mode carotid ultrasound was performed according to the standardized and validated scanning and reading protocols previously detailed [52]. Carotid intima-media thickness (cIMT) was measured per the consensus documents [53] using the automated computerized edge tracking software M’Ath (Intelligence in Medical Technologies, Inc., Paris, France) [54]. The cIMT protocols yield measurements of the distance between the lumen-intima and media-adventitia. Total cIMT is calculated as a composite measure of the means of the near and far wall IMT of all carotid sites (the common carotid artery, the internal carotid artery, and the bifurcation) from both sides of the neck. Atherosclerotic plaques were defined as the area of focal wall thickening that is 50% greater than the surrounding wall thickness. Peripheral blood was collected from each participant in K2E EDTA tubes by venipuncture. Plasma was isolated by centrifugation at 1000× *g* for 10 min at room temperature, collected, aliquoted, and stored at −80 °C until needed. 

### 4.2. Isolation & Characterization of Plasma EVs

As defined by the International Society for Extracellular Vesicles (ISEV) [39], EVs, or “extracellular vesicles,” are the lipid bilayer particles that are unable to replicate. EVs from human plasma were isolated by differential centrifugation, as previously described [55]. Briefly, 5 mL of plasma samples underwent a series of centrifugation (2000× *g* for 30 min followed by 12,000× *g* for 45 min) and ultra-centrifugation (110,000× *g* 2 h) steps at 4 °C. The pellet was resuspended in 15 mL of PBS, filtered through a 0.22 μm filter, and centrifuged at 110,000× *g* 70 min. The pellet was resuspended in 15 mL of PBS and centrifuged again at 110,000× *g* 70 min. Finally, the EV pellets were resuspended in 200 µL of PBS for downstream analyses. Protein concentration of EV samples was determined using the Pierce BCA Protein Assay Kit (ThermoFisher Scientific, Waltham, MA, USA). Nanoparticle tracking analyses (NTA) were used to examine the size distribution and particle number of EVs. For NTA, the isolated EVs were diluted 100 times and analyzed on the Nanosight LM10 system with Nanosight NTA 2.3 Analytical Software (Malvern Instruments Ltd., Malvern, UK). For this system, the range of diameter size detection limits is 10 nm~1000 nm. To evaluate the surface markers of EVs, EVs were incubated with anti-CD63 antibody-coated magnetic beads (Dynabeads) (ThermoFisher Scientific). The bead bound EVs were then incubated with rabbit anti-human antibodies against EV surface markers CD63, CD9, CD81, or Hsp70 (ExoAB Antibody Kit, System Biosciences, Palo Alto, CA, USA). The percentage of expression of each of these EV markers were assessed by flow cytometry (CytoFlex Flow Cytometer, Beckman Coulter, Brea, CA, USA). A sulfophosphovanilin colorimetric assay was used to assess the concentration of total lipids in EV preparations as previously described [56]. Briefly, 50 μL of EV suspension were mixed with 250 μL of 96% sulfuric acid in chloroform-pretreated tubes and incubated at 90 °C for 20 min with the tube lid open. Thereafter, 220 μL of samples were transferred into a 96-well polystyrene plate, cooled to room temperature, and incubated with 110 μL of 0.2 mg/mL vanillin in 17% phosphoric acid for 10 min at room temperature. Absorbance was measured at 540 nm using a spectrophotometric plate reader (Spectra Max M5, Molecular Devices, San Jose, CA, USA). 

### 4.3. EVs with Modified miR Contents

To generate EVs with modified miR contents, including tailored extracellular vesicles (TEVs) loaded with antagomirs for candidate EV-miRs, we used genetically modified bone marrow mesenchymal stromal cells (MSCs)—following an approach that was previously described by our group [12]. Lentiviral plasmids expressing antagomirs or miRs that are under investigation were purchased from System Biosciences). Lentiviral particles were generated using HEK 293T cells co-transfected with the antagomir/miR-expressing plasmid, the packaging plasmid psPAX2 (Addgene #12259), and the envelope glycoprotein expressing plasmid, pCMV-VSV-G (Addgene #8454). The culture supernatant containing the viral particles was collected 48 h post transfection and the viral particles were concentrated using the Lenti-X™ Concentrator (Takara, Kusatsu, Shiga, Japan) and quantified using the Lenti-X™ qRT-PCR Titration Kit (Takara). MSCs isolated from 3-month-old C57BL/6 *apoE^−/−^* mice were transduced with each construct to make EVs with modified miR content. After 48 h of transduction, cells were cultured in the presence of 3 µg/mL puromycin for 48 h. Surviving cells were washed with PBS and cultured in serum-free media for 24 h. EVs with modified miR contents were isolated from serum-free media using differential centrifugation as previously described [55]. Briefly, 80 to 90 mL of cell culture media underwent a series of centrifugation (300× *g* 10 min followed by 2000× *g* 10 min and 12,000× *g* 30 min) and ultra-centrifugation (100,000× *g* 70 min) steps at 4 °C. The pellet was carefully resuspended in 15 mL of PBS and centrifuged again 100,000× *g* 70 min. The final pellet was resuspended in 200 µL of PBS for further analysis. 

### 4.4. Animal Model

To determine the role of plasma EVs in mediating the effects of HIV infection on atherosclerosis, we used the atherosclerosis-prone *apoE*^−/−^ mouse model. All procedures were performed in accordance with a protocol approved by the Institutional Animal Care and Use Committee (IACUC) at the University of Miami (Protocol # 18-171-LF) and the Guide for the care and use of laboratory animals [57]. Animals were purchased from the Jackson Laboratory (Bar Harbor, ME, USA). Four-week-old male *apoE*^−/−^ mice were divided into four groups based on their weekly tail vein injection regimens: (1) circulating EVs from PLHIV (HIV^posEVs^), (2) PLHIV plasma depleted of EVs (HIV PL^depEVs^), (3) circulating EVs from HIV negative subjects (HIV^negEVs^) and (4) PBS. All animals from all groups were fed a high fat/high cholesterol diet. Animals were assigned to groups using simple randomization. There were five animals in each group. Inclusion criteria were sex, age, and overall good health assessment. The exclusion criteria during the course of the experiments were the presence of animal stress (behaviors such as hiding, pressing the head against cage walls), weight loss, or decreased food or water intake. No animals had to be excluded from the study. Mice were injected weekly for 12 weeks (12 injections total). The total protein amount of 500 µg was used for each injection in a total volume of 100 µL to maintain uniformity across all treatments. One week after the final injection, animals were sacrificed, and the aortas and bone marrow were collected. Euthanasia was conducted according to the AVMA Guidelines for the Euthanasia of Animals, with CO_2_ exposure as the primary method, followed by quick cervical dislocation done by trained personnel as a secondary method when needed. 

### 4.5. Atherosclerosis Measurement

The aorta was cut into thoracic and abdominal segments and embedded in Tissue-Tek OCT media. Multiple 8-μm sections were made for histological and immunofluorescent staining. Oil Red O (ORO), hematoxylin and eosin (H&E), and TUNEL staining were performed in alternating frozen sections, such that a survey of a substantial portion of each of the proximal thoracic and abdominal segments was obtained. In an effort to capture the aspect of the extension of the aortic lesions, more than one tissue section per animal was included in the final analysis. For H&E staining, aortic frozen sections were fixed for one minute with a 95% ethanol solution and stained with 0.1% Mayers hematoxylin for 1 min, then washed with water for 2 min. Next, slides were dipped in 0.5% Eosin solution for 30 s, washed in 95% ethanol solution and in absolute ethanol, followed by two xylene washes. For ORO staining, slides were fixed with 10% formalin for 10 min, rinsed with PBS for 1 min and then 60% Isopropanol for 15 s. Next, sections were stained with ORO Working Solution (Sigma Aldrich, St. Louis, MO, USA) for 1 min and rinsed in 60% isopropanol for 15 s, followed by 3 washes in PBS. Counterstaining for the nuclei was performed by mounting the sections with Prolong Diamond Antifade Mountant (DAPI, ThermoFisher Scientific). Tissue sections were viewed and photographed on a Zeiss Observer Z1 microscope. Atherosclerotic burden was calculated as the ratio of lesion area/total lumen area (ImageJ software, 1.46r, NIH). At least 5 ORO-staining histological sections for each segment were used to quantify the atherosclerosis burden. 

### 4.6. Quantification of Endothelial Colony-Forming Cells (ECFCs) from Humans

Circulating endothelial colony-forming cells, or ECFCs [7], formerly known as late endothelial progenitor cells, were quantified by colony-forming units (CFUs) using an established protocol [58]. Briefly, human peripheral blood mononuclear cells (PBMCs) were isolated using Ficoll-Paque reagent (Amersham Biosciences, Amersham, UK) from whole blood. Cells were washed twice in PBS with 2% FBS and plated on a 24-well plate coated with fibronectin at a density of 10^4^ cells/well in 500 µL endothelial basal medium-2 (EBM-2, Lonza, Basel, Switzerland) supplemented with EGM-2MV single aliquots and 10% FBS. In the first two days, the medium was changed daily to remove nonadherent cells and debris, and then every other day. After one week of cultivation, the cells were dissociated with trypsin and reseeded onto fibronectin-coated plates for further cultivation. After 2–3 weeks of culturing, colonies were stained with 6.0% glutaraldehyde and 0.5% crystal violet solution (Sigma-Aldrich) and counted using a stereomicroscope (OM2300S-GX4 3.5X-45X Zoom Stereo Boom Microscope, Omano, China). 

### 4.7. Isolation of lin^−^ BMCs from Mice

Mouse lin^−^ BMCs were isolated from bone marrow as previously described [13,28]. Briefly, femur and tibia bones were flushed with MEM media (Minimum Essential Medium, ThermoFisher Scientific) until translucent. Cell pellets from the flushed media were depleted of red blood cells (RBCs) using the Red Blood Cell Lysis Buffer (Millipore Sigma, Burlington, MA, USA). lin^−^ BMCs were isolated using the Mouse Lineage Cell Depletion Kit (Miltenyi Biotec Inc., Bergisch Gladbach, North Rhine-Westphalia, Germany). Isolated lin^−^ BMCs were cultured on fibronectin-coated (Sigma Aldrich) plates in EBM™-2 Endothelial Basal Medium-2 (Lonza) supplemented with EGM™-2 MV Microvascular Endothelial SingleQuots^TM^ Kit (Lonza) containing vascular endothelial growth factor (VEGF), fibroblast growth factor-2 (FGF2), epidermal growth factor (EGF), insulin like growth factor (IGF), ascorbic acid, hydrocortisone, gentamicin, amphotericin-B, and 20% fetal bovine serum (FBS). Cells at passages 3–7 were used for experimentation. 

### 4.8. Lipid Panel and C Reactive Protein ELISA Assays

Biochemical analyses were performed using a Vitros 5600 analyzer (Ortho Clinical Diagnostics, Raritan, NJ, USA). Total cholesterol, triglycerides, and HDL-C were directly determined, and LDL-C was calculated using the Friedewald equation [59] An enzyme-linked immunosorbent assay (ELISA) was used to quantify C-reactive protein in mouse serum according to the manufacturer’s protocol (Abcam, Cambridge, UK). 

### 4.9. Cell Migration “Scratch” Assay

To assess cell migration capacity, lin^−^ BMCs were seeded in 24-well plates (1 × 10^5^ cells per well). EV treatment was performed with a concentration of 500 µg of total EV protein per 1 mL of culture medium. Cell monolayers were scratched using a sterile pipette tip. The scratch areas were photographed and measured at 0 h, 24 h and 48 h post scratching using the EVOS FL Auto Fluorescence Inverted Microscope Imaging System (ThermoFisher Scientific). The area of scratches was analyzed with ImageJ software (NIH). 

### 4.10. Apoptosis Assay

Apoptosis in the aortic wall was detected and quantified using the TUNEL assay. TUNEL reactions were performed with the In Situ Cell Detection Kit, TMR red (Roche, Basel, Switzerland), according to the manufacturer’s protocol. In an effort to capture the true extent of the aortic lesions, the tissue was cut into multiple sections, with alternating sections being used for Oil Red O (ORO), hematoxylin and eosin (H&E), and TUNEL staining. Tissue from multiple mice was used and treated in the same manner.

### 4.11. Senescence Assay

Cellular senescence was evaluated by β-galactosidase assays. For cultured cells, cells were stained with the β-galactosidase staining kit (Cell Signaling Technology, Danvers, MA, USA) according to the manufacturer’s protocol. For frozen tissue sections, the Senescence Detection Kit (Abcam, Cambridge, MA, USA) was used according to the manufacturer’s protocol. Images were acquired in 10 random microscopic fields per sample at 10× magnifications using the EVOS FL Auto Fluorescence Inverted Microscope Imaging System (ThermoFisher Scientific). 

### 4.12. Cell Proliferation Assay

Cell proliferation was evaluated using the MTT (3-[4, 5-dimethylthiazol-2-yl]-2, 5-diphenyl tetrazolium bromide) assay. Briefly, lin^−^ BMCs or ECs were seeded in 96-well plates (5 × 10^3^ cells per well) and allowed to grow for 48 h in EBM^TM^-2 Endothelial Cell Growth Basal Medium-2 (Lonza) complete media. Fresh media containing 1 mg/mL of MTT (Millipore Sigma) was added to the culture wells, according to the manufacturer’s instructions. To quantify cell proliferation, MTT formazan crystals were eluted with isopropanol, and the absorbance of samples was assessed at 570 nm, using 690 nm as a reference wavelength, with a spectrophotometric ELISA plate reader (Spectra Max M5, Molecular Devices).

### 4.13. Small RNA Sequencing (sRNA-seq)

To identify EV-miRs that are associated with elevated atherosclerosis burden in PLHIV, we selected PLHIV with the largest plaque areas (N = 5) and HIV^neg^ subjects with no plaques (N = 5), representing the extremes of the disease spectrum. All individuals were male, non-smokers, and had an average age of 47 and 46 years, for the PLHIV and HIV^neg^ group, respectively. sRNA-seq of EVs from these samples was performed by System Biociences (SBI, Palo Alto, CA, USA). Briefly, EVs were isolated using the ExoQuick-ULTRA precipitation method, and a sRNA-seq library was prepared using the XRNA Exosome RNA-Seq Library Kit [60,61]. Sequencing was performed on the Illumina NextSeq with 1 × 75 bp single-end reads. On average, 5–10 million reads were generated for each library. After quality control (QC) using FastQC and end trimming, about 5 million reads per sample were mapped to the reference human genome using Bowtie [62]. SAMtools and Picard [63,64] were used for differential expression analyses, including read coverage, determination of small RNA abundance, and differential expression analysis across samples. Expression analyses were conducted based on normalized counts [counts per million (CPM)] of mature miR sequences. MiRs with a two or higher fold increase in the HIV^posEV^ group compared to controls were considered for further analysis. Of the upregulated miR sequences, the top 5 miRs with the highest expression level were prioritized for further validation by qRT-PCR. 

### 4.14. qRT-PCR Validation of Candidate EV-miRs

Small RNAs were extracted from EVs using the miRNeasy Mini Kit (Qiagen, Hilden, Germany). Reverse transcription and qPCR were performed using the specific miRNA primer and probe sets provided by the Taqman™ MicroRNA Assay kits (Thermo Fisher Scientific). The U6 small nucleolar RNA was used as an internal control [45,65,66,67]. TaqMan™ Gene Expression Assays were performed on the ABI PRISM 7300 system (Applied Biosystems, Waltham, MA, USA), and the relative expression was calculated using the 2^−ΔΔCt^ method. All values were normalized to U6 RNA levels. 

### 4.15. Western Blot

Briefly, cell and EV lysates (45 µg total protein amount) were loaded on 4–20% Tris-Glycine Gels (ThermoFisher Scientific) and transferred to Polyvinylidene Difluoride (PVDF) membranes (Sigma Aldrich). Membranes were blocked and subsequently exposed to primary antibodies (primary antibodies used: Hmga2 Rabbit Ab, Cell Signaling #5269S; GAPDH Rabbit Ab, Cell Signaling #2118S, ApoA1, Thermo Fisher Scientific, 701239; ApoB, Thermo Fisher Scientific, MA5-35458). Membranes were washed and exposed to secondary antibodies (anti-rabbit HRP, Millipore, AQ132P, anti-mouse HRP, Cell Signaling, #7076). Membranes were developed using SuperSignal West Pico Chemiluminescent Substrate (ThermoFisher Scientific).

### 4.16. Ingenuity Pathway Analyses (IPA)

IPA’s Target Analysis Tool was utilized to identify gene targets of candidate EV-miRs. IPA’s “Disease and Functions” tool was used to identify the diseases or cellular functions that were significantly enriched with the target genes of the miRs of interest. A FDR *p*-value < 0.01 was considered significant for the analyses. The *p*-value for each “Disease/Function category” is provided as a range composed of multiple p-values for each of the subcategories within the larger disease or function category (e.g., the cardiac enlargement category is made up of the categories of enlargement, ventricular hypertrophy, cardiomegaly, etc., each with its own *p*-value). 

### 4.17. Transduction of Mouse lin^−^ BMCs

To dissect the molecular mechanisms underlying the EV-miR effects, we focused on High Mobility Group AT-Hook 2 (*Hmga2*), a known target of let-7b-5p and a gene shown to regulate the senescence of neural stem cells and lin^−^ BMCs [28,30]. To that end, we conducted a series of transduction experiments in lin^−^ BMCs isolated from young wild type mice overexpressing either wild-type *Hmga2* (wt*Hmga2*) or *Hmga2* lacking the 3′UTR (*Hmga2*-3′UTRdel, therefore resistant to miR regulation). Lentiviral transduction and cell selection were performed as described above.

### 4.18. In Vitro EV Treatment

Cells were seeded in 24-well plates (1 × 10^5^ cells per well). EBM™-2 Endothelial Basal Medium-2 (Lonza) supplemented with EGM™-2 MV Micro-vascular Endothelial SingleQuotsTM Kit (Lonza) was mixed with EVs at a concentration of 500 µg protein per mL of culture media for 5 days (media with EVs was changed every 2–3 days). 

### 4.19. Statistical Analyses

Kolmogorov-Smirnov and Shapiro-Wilks normality tests were used to assess normal distribution. The Kruskal-Wallis test was used to compare differences between treatment groups and the control group for non-parametric variables, followed by the Dunn’s Multiple Comparison test. For continuous variables, Student’s *t* test (*t* test) and analysis of variance (ANOVA) were used to compare differences between two groups and among ≥3 different groups, respectively. For the animal model, a total sample of 20 subjects achieves 80% power to detect differences with a 0.05 significance level and a relatively large effect size of 0.84. Although we have a relatively small sample, based on our preliminary study with an effect size above 2.5, we have enough power to detect the difference across 4 groups. Statistical analysis was performed using GraphPad Prism 7 (GraphPad, San Diego, CA, USA). *p* < 0.05 was considered statistically significant. For sRNA-seq, expression statistics were calculated and visualized using R 3.5.2.

## Figures and Tables

**Figure 1 ijms-24-07567-f001:**
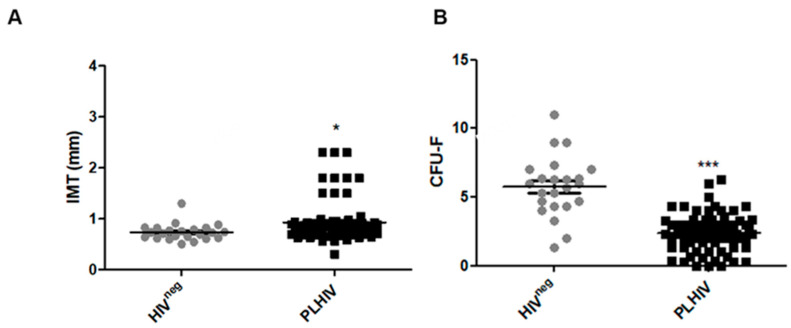
PLHIV have increased atherosclerosis and decreased ECFCs levels. (**A**). cIMT in PLHIV is higher than that in HIV^neg^ subjects (* *p* < 0.05). (**B**). Circulating ECFC levels as determined by CFUs in PLHIV are lower than in HIV^neg^ individuals (*** *p* < 0.001). PLHIV (N = 74) and HIV^neg^ subjects (N = 23). Statistical test used: student’s *t*-test.

**Figure 2 ijms-24-07567-f002:**
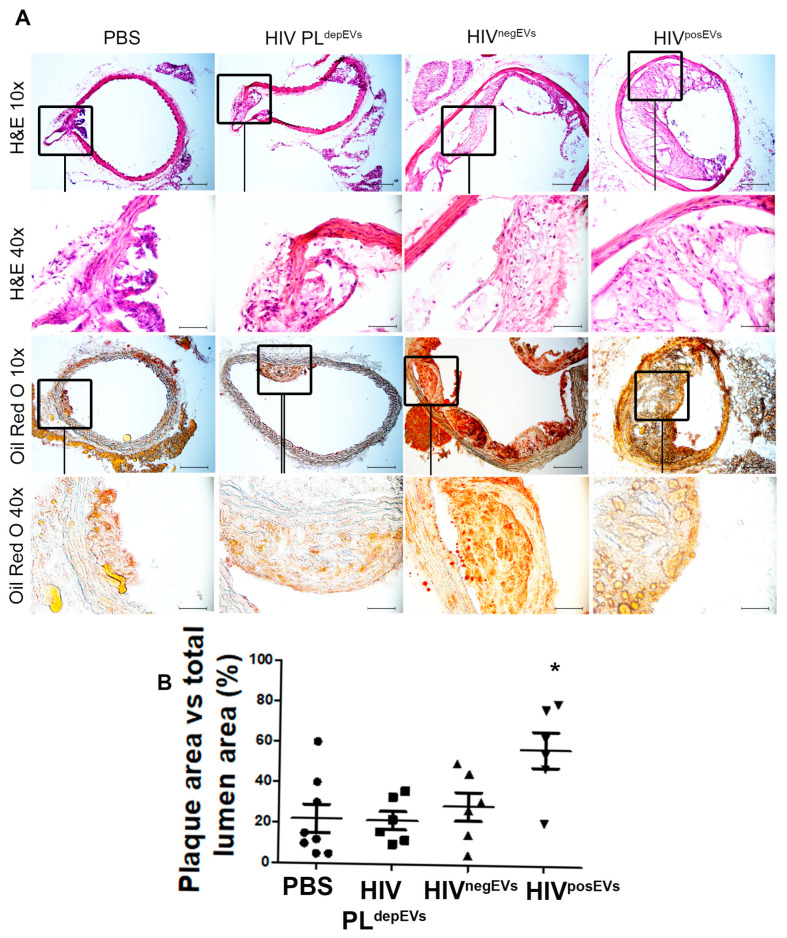
HIV^posEVs^ increase atherosclerosis burden in *apoE*^−/−^ mice. (**A**). Representative images of H&E and Oil Red O staining of aortic sections from mice treated with HIV^posEVs^, HIV PL^depEVs^, HIV^negEVs^ or PBS. Mice treated with HIV^posEVs^ have the most severe atherosclerotic lesions. (**B**). The ratio of plaque area/total lumen area in the four treatment groups (* *p* < 0.05). N = 5 animals per group. Statistical tests used: Kruskal–Wallis and Dunn’s Multiple Comparison Test as a post test. Scale bars: 10×, 200 µm; 40×, 50 µm.

**Figure 3 ijms-24-07567-f003:**
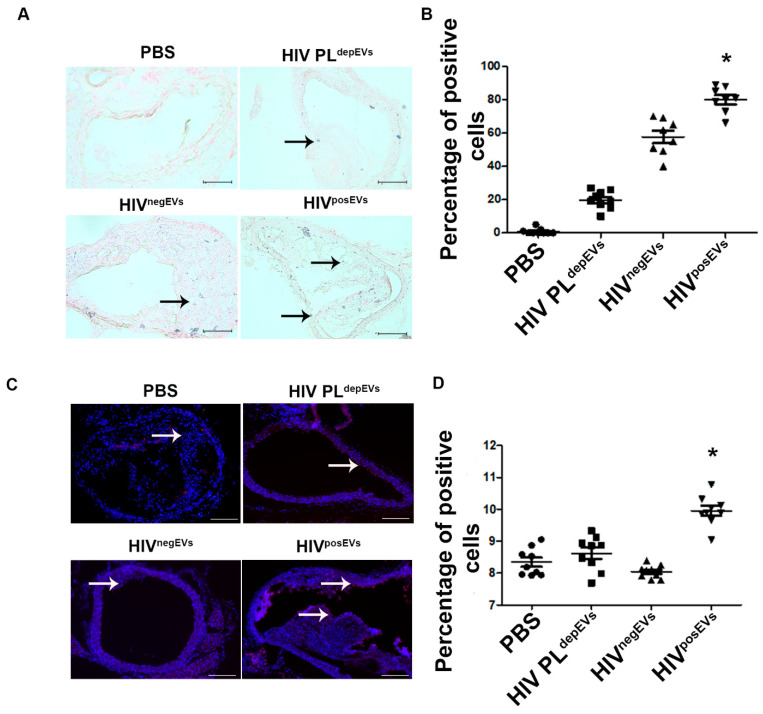
HIV^posEVs^ treatment results in impaired vascular repair/rejuvenation in vivo. (**A**,**B**). β-gal staining shows a greater number of senescent cells (black arrows show examples of positive cells, stained in blue) in the aortas of *apoE*^−/−^ mice treated with HIV^posEVs^. (**C**,**D**). The TUNEL assay shows more apoptotic cells (examples of positive cells are pointed by white arrows. Positive cells are red) in the vascular wall of aorta from *apoE*^−/−^ mice treated with HIV^posEVs^ than mice treated with HIV PL^depEVs^, HIV^negEVs^ and PBS. Each group had five animals (* *p* < 0.05). Statistical tests used: Kruskal–Wallis and Dunn’s Multiple Comparison Test as a post test. Enlargements of the representative images are available in Appendix A. Scale bars: 200 µm.

**Figure 4 ijms-24-07567-f004:**
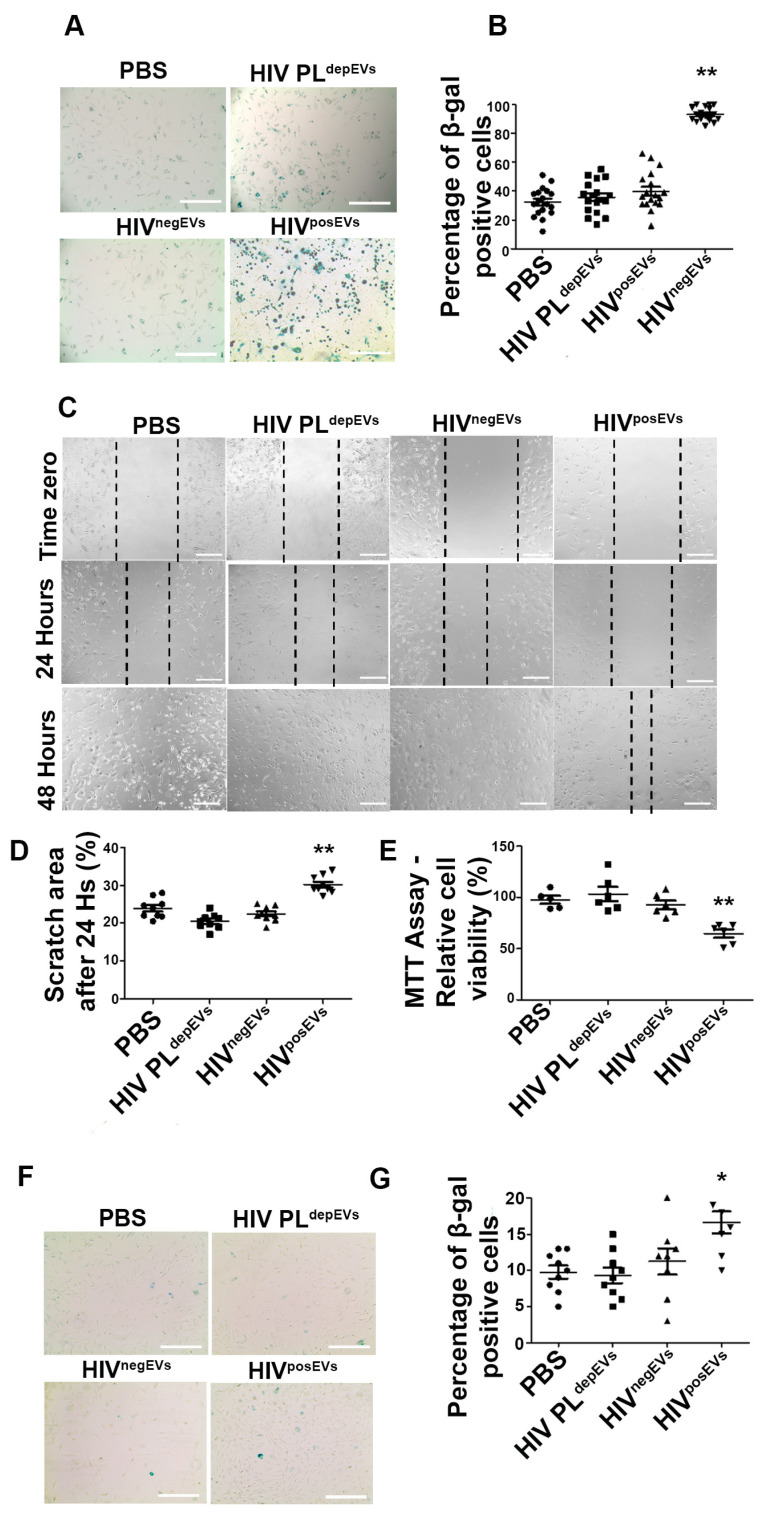
HIV^posEVs^ promote lin^−^ BMC senescence and impair lin^−^ BMC functions in vivo and in vitro. (**A**,**B**). Lin^−^ BMCs isolated from HIV^posEVs^-treated *apoE*^−/−^ mice show increased β-gal staining, as compared with lin^−^ BMCs from mice treated with HIV PL^depEVs^, HIV^negEVs^ or PBS. (**C**,**D**). Lin^−^ BMCs from HIV^posEVs^-treated *apoE*^−/−^ mice show slower migration than cells from mice treated with HIV PL^depEVs^, HIV^negEVs^ or PBS. Dashed lines in panel C mark the edges of the scrach in the cell monolayer. (**E**). Lin^−^ BMCs from HIV^posEVs^-treated *apoE*^−/−^ mice display reduced proliferation in comparison with control treatments. (**F**,**G**). Lin^−^ BMCs isolated from 3-week-old untreated *apoE*^−/−^ mice were treated in vitro with HIV^posEVs^ for 96 h, which resulted in increased percentage of senescent cells (β-gal staining) than lin^−^ BMCs treated with HIV PL^depEVs^, HIV^negEVs^ or PBS. For the in vivo experiment, each group had five animals. For the in vitro experiment, all experiments were performed in triplicate and repeated thrice (* *p* < 0.05, ** *p* < 0.001). Statistical tests used: Kruskal–Wallis and Dunn’s Multiple Comparison Test as a post test. Scale bars: (**A**,**F**): 200 µm; (**C**): 400 µm.

**Figure 5 ijms-24-07567-f005:**
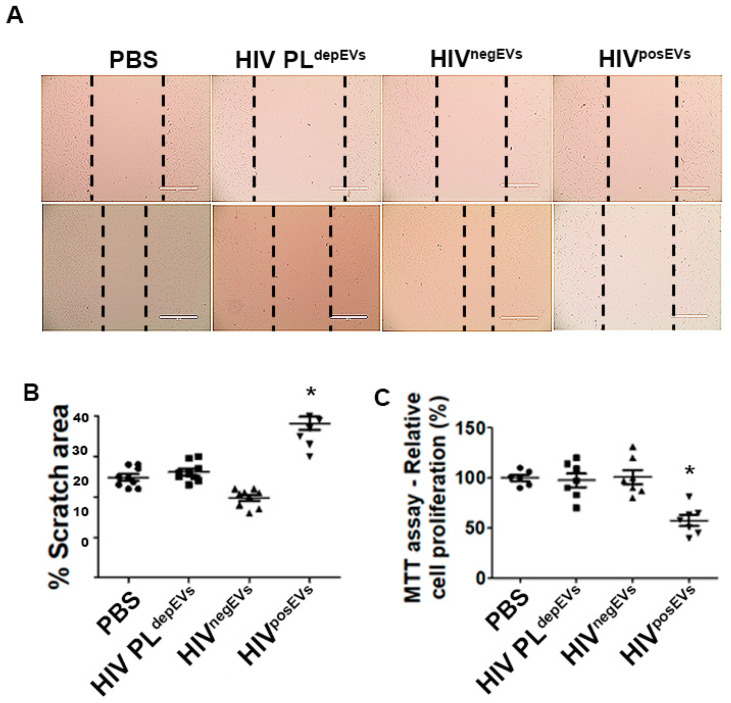
HIV^posEVs^ treatment results in impaired migration and diminish proliferation in ECs in vitro. Mouse aortic ECs isolated from 3-week-old *apoE*^−/−^ mice exposed to HIV^posEVs^ show slower migration (**A**,**B**) and lower cell proliferation after 96 h of exposure (**C**), compared with HIV PL^depEVs^, HIV^negEVs^ and PBS. Dashed lines in panel C mark the edges of the scrach in the cell monolayer**.** The experiments were performed in triplicate and repeated thrice (* *p* < 0.05). Statistical tests used: Kruskal–Wallis and Dunn’s Multiple Comparison Test as a post test. Scale bars: 400 µm.

**Figure 6 ijms-24-07567-f006:**
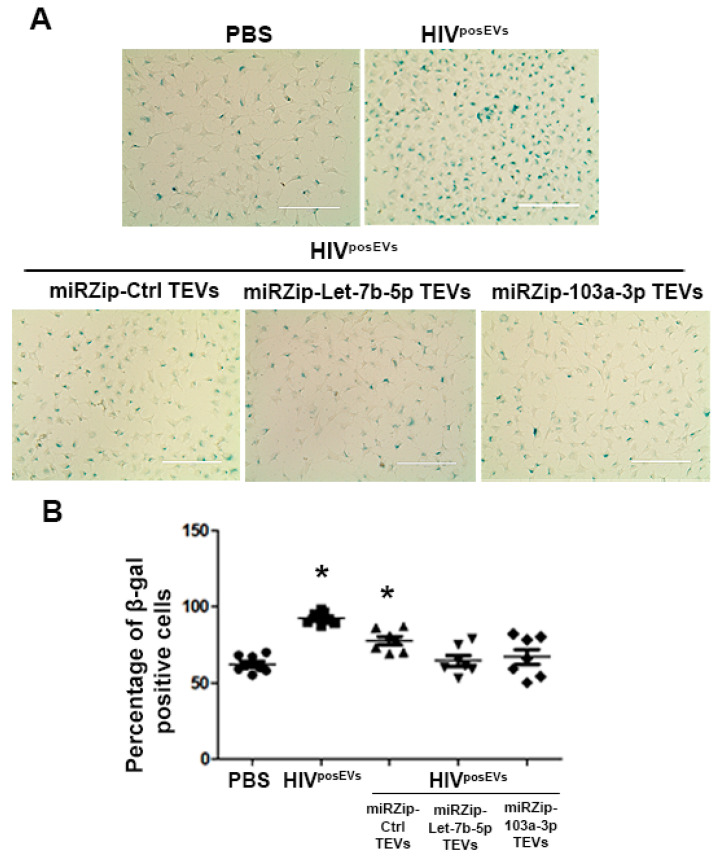
Tailored EVs containing antagomirs for let-7b-5p and miR-103a-3p abrogate the effects of HIV^posEVs^ in lin^−^ BMC senescence in vitro. (**A**). Representative images of lin^−^ BMCs incubated with HIV^posEVs^ or HIV^posEVs^ plus miRZip-Ctr show increased percentages of β-gal positivity. By contrast, lin^−^ BMCs exposed to miRZip-103a-3pTEVs and miRZip-let-7b-5p TEVs combined with HIV^posEVs^ (1:1 ratio) for 96 h show similar percentages of β-gal positivity relative to cells exposed to PBS. (**B**). Quantification analysis shows the statistically significant changes among these treatment groups (* *p* < 0.05 compared to PBS treatment). The experiments were performed in triplicate and repeated thrice. Statistical tests used: Kruskal–Wallis and Dunn’s Multiple Comparison Test as a post test. Scale bars: 200 µm.

**Figure 7 ijms-24-07567-f007:**
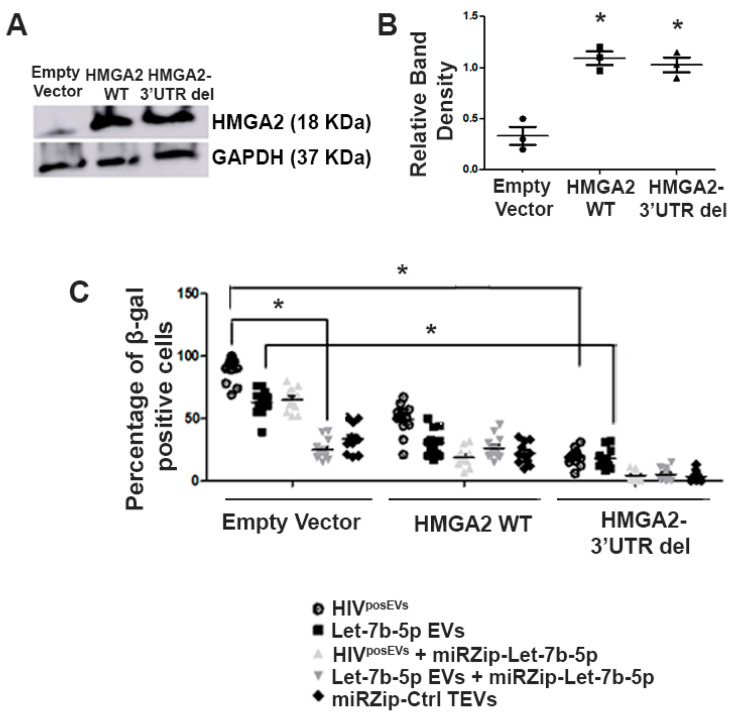
Hmga2 with 3′UTR deletion attenuates HIV^posEVs^ effect in lin^−^ BMC senescence. (**A**,**B**). Expression and quantification of Hmga2 in lin^−^ BMCs transduced with different expression vectors. (**C**). Cellular senescence is measured by β-gal staining. Let-7b EVs partially recapitulate the effects of HIV^posEVs^ on lin^−^ BMCs. In Hmga2-3′UTRdel-overexpressing lin^−^ BMCs, the effects of HIV^posEVs^ on β-gal expression were partially blocked, similar to co-incubation with miRZip-let-7b-5p TEVs, whereas Hmga2-3′UTRdel overexpression almost completely blocked the effects of let-7b EVs. Furthermore, wt Hmga2 overexpression modestly attenuated the effects of HIV^posEVs^ and let-7b EVs. The experiments were performed in triplicate and repeated thrice. Statistical tests used: Kruskal–Wallis and Dunn’s Multiple Comparison Test as a post test. Representative images for the β-gal experiment are available in the Appendix A (* *p* < 0.05).

## Data Availability

The data presented in this study are available on request from the corresponding author.

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
