# Peer review of "HIV Promotes Atherosclerosis via Circulating Extracellular Vesicle MicroRNAs"

_ijms, 2023, doi:10.3390/ijms24087567_

Round 1
Reviewer 1 Report
Dear Editor,
I carefully read the manuscript entitled "HIV Promotes Atherosclerosis via Circulating Extracellular Vesicle microRNAs", that is a very well written and interesting review article.
My comments and suggestions for the authors are the following:
- In the Background, the authors should consider to refer to doi: 10.1016/j.atherosclerosis.2022.06.001.
- Lines 231, 232: Did the authors refer to LDL-C and HDL-C? Usually, LDL and HDL refer to particles instead.
- The main flaw of the study is the small sample size. The authors should include among the statistical methods information as regards how the sample size was measured (i.e. How was the power analysis performed?).
- Line 324: The authors should specify how the normal distribution of the variables was assessed.
- The limitations of the study should be further and more deeply discussed.
- In the Discussion, the authors should consider to refer to doi: 10.3390/ijms23052504 and doi: 10.1016/j.jacc.2017.05.012.
Reviewer 2 Report
Overall
The manuscript “HIV Promotes Atherosclerosis via Circulating Extracellular Vesicle microRNAs” by Andrea Da Fonseca Ferreira and co-authors presents detailed and thoughtful experiments to elucidate the role of extracellular vesicles (EVs) found in plasma of people living with HIV (PLHIV) in the promotion of cardiovascular disease, particularly the role of miRNA contents of EVs in functional aspects of atherosclerosis. They used well defined groups of human subjects, as well as in vitro experiments and well-established atherogenic mouse model, as platforms to evaluate their hypotheses. Their work is detailed and technical, but well explained in the text and all findings are supported by the experiments, text and figures. The figures could be larger and more clearly labeled in some cases, but this is a minor point that should be easy for the authors to address. This work builds on their previous work and the work of others, presents cellular mechanisms of the persistent influence of HIV on the cardiovascular system in PLHIV even with undetectable viral loads, and provides a foundation for future studies including interventions and biomarker studies. I recommend the authors address the minor points below, and that the journal publish the article after the minor points are addressed.
Introduction
Lines 40-41: Consider including that PLHIV have elevated rates of non-AIDS-related comorbidities, including CVD. Add references to support the statement if made.
Lines 41-43: “The persistent low-level viremia, detectable only by ultrasensitive PCR, coupled with the chronic state of immune activation and inflammation, represent the most significant factor contributing to the elevated risk for the development of CVD.” These are two factors, while related, are still two factors. This statement needs references.
Lines 71-72: “As such, EVs serve as important vehicles mediating cell-to-cell communication.” The authors describe the EV cargo and secretion, but not the absorption of EVs. Consider adding a sentence about the absorption of EVs and use of their cargo by other cells to put this sentence/idea in context.
Lines 74-75: Consider adding an additional statement and references about continued presence of HIV proteins in plasma EVs even while plasma virus is undetectable on cART. (29364927, 36417867, 31344124, others)
Methods
Lines 299-300: This sentence has some grammatical problems that would be easy to fix.
Lines 319-320: 2.18 In vitro EV treatment needs to be expanded with more information including the cell density when plated, kind of plate used, culture media used, etc. If the exact conditions used are described previously, please refer to the section where that information is found. Incubating cells for 5 days in the same media seems like a very long time, so if the cells were passaged or the media was changed, please give details.
Results
Overall: The “HIV PlasmadepEVs” group label makes for a large amount of negative white space around some of the figures, particularly dot plots. If Plasma was perhaps abbreviated “HIV PLdepEVs” or just removed “HIVdepEVs”, and the group defined as such in the text, the negative space around the figures can be reduced and the size of the dot plots increased, which would enhance clarity of the data in the figures.
Figures 2 and 3: Legend says there are 5 animals per group but 2B and 3B/3D look like they have more than 5 symbols for each group. Please explain.
Figure 3, panels B and D: It is hard to distinguish the asterisk indicating significance, particularly for panel B. These parts of the figure be larger to enhance clarity, or asterisks could be moved to the top of the figure or at least further away from the experimental values to enhance clarity.
Figure 3, panels A and C: The images and arrows are very hard to interpret at the current size. The colors are not distinct. It is unclear what the arrows are indicating – is it an example of a stained cell, or the only stained cell in the image? The legend seems to indicate that the arrows are showing the only senescent/apoptotic cells, which I don’t think is the case. Perhaps these images can be larger, and included in a supplementary figure if the authors are worried about using space in the main text to include larger easier to read images.
Figure 4: The dot plots need to be larger for clarity (at least the size of 5B&5C). The methods of putting the four images in a 2x2 block worked better, rather than displayed horizontally (compare 4A with 3A).
Figure 5C: Was significance found between HIVposEVs and other groups? There appears to be a large difference in the means but no asterisk is visible.
Section 3.7 and Supp Figure 3: The Venn diagram is informative, the text below (3A) is unnecessary as it is in the text, or could be included in the legend as additional information if the authors prefer to reiterate. 3B is unnecessary, as it is well explained in the text. The data shown in the rest of the figure could be increased in size to improve clarity.
Figure 6B: Is the horizontal bar over the miRZips on the x-axis supposed to continue across all three? Please make it more clear.
Figure 7: The microscope images take up a lot of space, but are not clear to interpret, particularly for statistically supported trends. Perhaps these images could be in a supplemental figure, and figure 7 could focus on the western and the dot-plots. Panel D is particularly interesting, and would benefit from being larger. Consider harmonizing the font sizes in Panel D, as the y-axis label is quite large compared to the x-axis. The labels on the symbols in the legend could use a larger font and may be better positioned under the dot plot rather than to the right.
